# Levodopa-Carbidopa Intestinal Gel Improves Symptoms of Orthostatic Hypotension in Patients with Parkinson’s Disease—Prospective Pilot Interventional Study

**DOI:** 10.3390/jpm12050718

**Published:** 2022-04-29

**Authors:** Simona Stanková, Igor Straka, Zuzana Košutzká, Peter Valkovič, Michal Minár

**Affiliations:** 12nd Department of Neurology, Faculty of Medicine, University Hospital in Bratislava, Comenius University in Bratislava, 833 05 Bratislava, Slovakia; stankova.simona@gmail.com (S.S.); straka0105@gmail.com (I.S.); zuzanakosutzka@gmail.com (Z.K.); peter.valkovic@gmail.com (P.V.); 2Institute of Normal and Pathological Physiology, Centre of Experimental Medicine, Slovak Academy of Sciences, 813 71 Bratislava, Slovakia

**Keywords:** Parkinson’s disease, levodopa-carbidopa intestinal gel, autonomic dysfunction, orthostatic hypotension, fluctuations

## Abstract

Parkinson’s disease (PD) is currently considered progressive neurodegeneration of both the central and peripheral nervous systems. Widespread neuropathological changes lead to a complex clinical presentation with typical motor (hypokinesia, tremor, and rigidity) and various nonmotor symptoms. Orthostatic hypotension is one of the most disabling nonmotor features contributing to increased morbidity and mortality and decreased quality of life (QoL). Our study aimed to disclose the effect of a continuous infusion of levodopa-carbidopa intestinal gel (LCIG) on symptoms of orthostatic hypotension. Nine patients indicated for LCIG and eight matched patients on optimized medical treatment (OMT) were examined with scales for orthostatic symptoms (SCOPA-AUT), nonmotor symptoms and motor fluctuations (MDS-UPDRS), and QoL (PDQ39) at both baseline and after six months. The scores of “light-headedness after standing” and “fainting” decreased in the LCIG group compared to the OMT group. Treatment with LCIG was associated with a significantly higher decrease in the score of “light-headedness after standing”. Change in the PDQ39 correlated positively with fluctuation improvement and with change in the scores of both “light-headedness” and “fainting”. LCIG treatment improved symptoms of orthostatic hypotension in patients with PD mainly by a reduction in motor complications. Decreased severity in both motor and nonmotor fluctuations was connected also with improved QoL. Continuous treatment with LCIG should be considered not only in the case of severe motor fluctuation but also in patients with nonmotor fluctuations responsive to dopaminergic treatment.

## 1. Introduction

Nonmotor symptoms (NMSs) occur in all patients with Parkinson’s disease (PD) and are a significant determinant of reduced quality of life (QoL) [1]. They arise due to the dysfunction of several neurotransmitter systems, including dopaminergic, noradrenergic, cholinergic, and serotonergic. However, some NMSs are still responsive to dopaminergic treatment. Moreover, higher doses of levodopa may lead to the onset or worsening of some of the NMSs [2].

Blood pressure fluctuations in levodopa-treated patients with PD have been known for decades [3,4]. In general, cardiovascular autonomic dysfunction is a part of the spectrum of NMSs in PD, and it includes orthostatic hypotension, postprandial hypotension, supine hypertension, and nondipping (a lack of nocturnal blood pressure fall).

Orthostatic hypotension is the most disabling feature of autonomic dysfunction [5]. It is defined as a sustained reduction of at least 20 mm Hg in systolic blood pressure or 10 mm Hg in diastolic blood pressure after 3 min of standing (or head-up tilt table testing) [6]. Some patients are asymptomatic, others might have unrecognized cognitive impairment due to decreased brain perfusion, and many patients can suffer from syncope with consecutive falls, which increases their overall morbidity or mortality and leads to a higher number of hospital admissions [7]. Because of its variable presentation, real data about the prevalence and modifying factors of orthostatic hypotension in PD are not well-established, but according to a systematic review and meta-analysis, the prevalence of orthostatic hypotension in PD is 30% [8,9].

The pathomechanism of orthostatic hypotension in PD is primarily neurogenic and involves inadequate neurocirculatory responses to postural change due to baroreflex failure and impaired release of norepinephrine [10]. Non-neurogenic causes of orthostatic hypotension include reduced intravascular volume (in PD, affected also by impaired thermoregulation and hyperhidrosis), cardiac comorbidities, and drug-induced hypotension [5]. Moreover, orthostatic hypotension leads to increased blood pressure variability and may contribute to nondipping or reverse dipping, which are risk factors for negative cardiovascular outcomes [11].

Advanced PD is associated with motor and nonmotor fluctuations. They are characterized as diurnal changes in “ON” time (i.e., time of favorable effect of dopaminergic treatment with good motor performance) and “OFF” time (the time when PD medication wears off and has an insufficient effect on motor function). They arise as a side effect of the long-term use of oral levodopa preparations [12] and lead to more complicated PD management, an increased need for nursing care, and impaired QoL [13]. In cases of motor and nonmotor fluctuations that can no longer be managed with optimized medical treatment (OMT), advanced therapeutic methods are considered, including levodopa-carbidopa intestinal gel (LCIG) treatment. The advantage of LCIG is the continuous administration of levodopa via a percutaneous endoscopic gastrostomy with a jejunal extension (PEG-J) throughout the day, ensuring a constant plasma level of levodopa, which is reflected in the significant reduction in fluctuations.

According to the results of the multinational observational GLORIA Registry, LCIG monotherapy led to significant reductions in “OFF” time, “ON” time with dyskinesia (involuntary movements due to dopaminergic overdose) and dyskinesia-related disability and pain, and significant improvements in measures of motor symptoms, activities of daily living, QoL, and NMSs [14]. Various selected observational studies and clinical trial studies have supported the use of LCIG in improving NMSs, especially mood, cognition, sleep, gastrointestinal, and urinary symptoms in PD patients [15]. The evidence of an impact on cardiovascular symptoms is scarce.

The aim of our study was to disclose the effect of LCIG treatment on symptoms of orthostatic hypotension in patients with PD and its possible connection with fluctuations.

## 2. Material and Methods

### 2.1. Methods

This was a prospective interventional study at the Movement Disorders Centre of the Second Department of Neurology, Derer’s University Hospital, Comenius University in Bratislava, Bratislava, Slovakia. Enrollment began in January 2021, and the study was completed in January 2022. The protocol was approved by the Ethical Committee of the Derer’s University Hospital, Bratislava, Slovakia, under approval number 13/2021. Written informed consent was received from all the subjects prior to inclusion in the study.

Relevant demographic and clinical data (age, disease duration, modified Hoehn and Yahr score, and antiparkinsonian medication) were collected.

For the purpose of our study, we used the following scales:Scales for Outcomes in Parkinson’s Disease—Autonomic Dysfunction (SCOPA-AUT) [16]. For the purposes of this study, we used following questions:▪Question 15: “In the past month, did you become light-headed after standing for some time?”▪Question 16: “Have you fainted in the past 6 months?”Movement Disorders Society—Unified Parkinson’s Disease Rating Scale (MDS-UPDRS) [17]. For the purposes of this study, we used following questions:○Part I—Nonmotor Symptoms▪1.12 “Lightheadedness on standing”○Part IV—Motor Complications▪Total score▪4.1 Time spent with dyskinesias▪4.2 Functional impact of dyskinesias4.3 Time spent in the “OFF” state▪4.4 Functional impact of fluctuations▪4.5 Complexity of motor fluctuations▪4.6 Painful “OFF” state dystoniaThe Parkinson’s Disease Questionnaire (PDQ-39) [18,19].

All the scales were administrated by trained investigators; higher scores indicated more severe symptoms. The patients were examined in their best ON state.

Treatment dose adjustments were allowed, but patients had to be on a stable dose for four weeks prior to evaluation at both of the two timepoints:(1)A baseline visit for patients indicated for LCIG before treatment initiation;(2)After six months.

For a review of literature focusing on orthostatic hypotension in PD, a publication search was undertaken using PubMed database and relevant search terms. The articles were screened for suitability and data relevance.

### 2.2. Participants

Inclusion criteria:-Idiopathic PD according to the MDS Clinical Diagnostic Criteria for PD [20];-Willingness to participate in a six-month follow-up;-Ability to complete all the questionnaires (the assistance of caregiver was allowed).

Exclusion criteria:-Montreal cognitive assessment (MOCA) below 10 (severe dementia);-Severe cardiovascular comorbidity other than arterial hypertension.

We enrolled:(1)Nine (*n* = 9) consecutive patients in advanced-stage PD according to the Delphi expert consensus panel “5-2-1 criteria” taking oral levodopa at least five times per day with at least 2 h of the day with ‘Off’ symptoms or at least 1 h of the day with troublesome dyskinesia [21] in which oral regimens no longer adequately managed PD symptoms and were indicated for continuous LCIG treatment (LCIG group);(2)Eight patients with idiopathic PD not fulfilling 5-2-1 criteria treated with OMT and matched by age, gender, and PD duration with patients indicated for LCIG (OMT group).

The male gender represented 87.5% in the OMT group and 88.9% in the LCIG group. The median of the Hoehn and Yahr score (used to describe the symptom progression or stage of PD) [22] was 2.5 for the OMT group and 3.0 for the LCIG group (IQR of 0.125 and 0.000, respectively).

Optimized medical treatment (OMT) included any combination of oral preparations of levodopa (with carbidopa or benserazide), entacapone, rasagiline, amantadine, pramipexol, and ropinirole, as well as rotigotine in a transdermal patch, in order to achieve a satisfactory motor state.

The baseline total levodopa equivalent daily dose was higher in the LCIG group (*p* = 0.048), and the equivalent daily dose of dopamine agonists was not significantly different between the LCIG and OMT groups (0.248). In the LCIG group, 44% (4 out of 9) of the patients were treated with dopamine agonists ((DA) 3 with rotigotine and 1 with pramipexol); after switching to LCIG, 5 patients were on monotherapy with LCIG, and 4 patients had comedication with rotigotine.

In the OMT group, 75% (6 out of 8) of the patients were treated by DA—3 with pramipexol and 3 with ropinirole—this medication did not change during the follow-up.

In both groups, four patients were treated with antihypertensive drugs at stable doses for four weeks prior to evaluations.

The demographic and basic clinical data at the baseline are summarized in Table 1.

### 2.3. Statistical Analysis

We determined the required minimum sample size on the basis of an a priori Power analysis [23] using the G * Power 3 program [24]. The data were analyzed by JASP Team (2021) JASP (Version 0.16) computer software. Descriptive statistics were used to evaluate the demographic and clinical data. For a normality evaluation, individual variables were first tested with the Lilliefors modification of the Kolmogorov–Smirnov test. Continuous parametric data satisfying normal distribution were described as mean ± standard deviation. If the data were non-normally distributed, they were described as median values with their corresponding interquartile. Categorical parametric data were presented as percentages.

We used repeated measure ANOVA to compare the effect of different treatment methods on the relevant outcomes. We calculated the difference between groups (type of treatment), the difference between timepoints, and the interaction between the type of treatment and both timepoints. The effects were considered small (η^2^ < 0.01), medium (η^2^ < 0.06), and large (η^2^ < 0.14). The Spearman correlation coefficient was used in the correlation analysis. The correlational analysis was followed by a regression analysis with the aim of identification of the most important predictors of change in relevant parameters. The magnitude of the indirect effect of motor complications was assessed by mediation analysis; the significance of the indirect effect was tested using a bootstrapping method on 1000 samples.

## 3. Results

### 3.1. Changes in the Symptoms of Orthostatic Hypotension

The score of SCOPA-AUT 15 (“In the past month, did you become light-headed after standing for some time?”) decreased in the LCIG group after six months compared to the OMT group (F = 8.000, *p* = 0.030, η^2^ = 0.082, medium effect size, Figure 1A).

The score of SCOPA-AUT 16 (“Have you fainted in the past 6 months?”) decreased in the LCIG group after six months compared to the OMT group (F = 4.200, *p* = 0.080, η^2^ = 0.125, medium effect size, Figure 1B).

The score of MDS-UPDRS, Part I, question 12 (“Lightheadedness on standing”) decreased in the LCIG group after six months compared to the OMT group (F = 3.316, *p* = 0.111, η^2^ = 0.052, small effect size, Figure 1C). All the data are shown in Table 2.

A higher score of SCOPA-AUT 15 (“In the past month, did you become light-headed after standing for some time?”) predicted a higher score of SCOPA-AUT 16 (“Have you fainted in the past 6 months?”) with the results of β = 0.222, *p* = 0.020, and 95% CI 0.041–0.404.

### 3.2. Relationship with PD Characteristics

Controlling for age and gender, the relative change in SCOPA-AUT 15 (“In the past month, did you become light-headed after standing for some time?”) after six months correlated negatively with the H and Y score (r_s_ = 0.564, *p* = 0.028) and positively with the MDS-UPDRS part IV total score (r_s_ = 0.628, *p* = 0.012) of question 2 (“Functional impact of dyskinesias” (r_s_ = 0.674, *p* = 0.006)) and question 4 (“Functional impact of fluctuations” (r_s_ = 0.640, *p* = 0.010)).

The relative change in SCOPA-AUT 16 (“Have you fainted in the past 6 months?”) after six months correlated positively with the MDS-UPDRS part IV question 1 (time spent with dyskinesias (r_s_ = 0.515, *p* = 0.050)).

Treatment with LCIG was associated with a significantly higher decrease in the SCOPA-AUT 15 score (β = −0.708, *p* = 0.012, and 95% CI (−1.235)–(−0.182)), but not after controlling for the mediator (MDS-UPDRS IV score; β = 0.026, *p* = 0.891, and 95% CI (−0.343)–0.395). These results indicated a significant indirect effect.

There was no significant correlation between MDS-UPDRS 1.12 (“Lightheadedness on standing”) and PD characteristics.

There was no significant correlation between the symptoms of orthostatic hypotension and PD duration or LEDD (nonsignificant data are not shown).

The changes in all of the subscores of the MDS-UPDRS, Part IV (motor complications), are shown in Table 3.

### 3.3. Relationship with QoL

The PDQ-39 total score decreased in the LCIG group after six months compared to the OMT group (F = 2.979, *p* = 0.128, η^2^ = 0.06, medium effect size, Figure 2).

Change in the PDQ39 correlated positively with fluctuation improvement (r_s_ = 0.554, *p* = 0.032, Figure 3) and with change in the scores of both SCOPA-AUT 15 (“In the past month, did you become light-headed after standing for some time?”) and SCOPA-AUT 16 (“Have you fainted in the past 6 months?”)—r_s_ = 0.622, *p* = 0.01; and r_s_ = 0.598, *p* = 0.019, respectively.

## 4. Discussion

Our data disclosed that symptoms of orthostatic hypotension were significantly alleviated in patients after six months of LCIG treatment compared to those on OMT. However, reviewing the previous literature, the data on the effect of LCIG on the symptoms of orthostatic hypotension are inconsistent. According to some studies, LCIG has significantly improved the score of the cardiovascular domain of the Non-Motor Symptoms Scale for Parkinson’s Disease (NMSS), which also focuses on the symptoms of orthostatic hypotension [25,26,27]. Other studies have reported a nonsignificant trend for improvement in the cardiovascular domain of NMSS [28,29]. According to a multicentric Barcelona Registry study examining long-term response to LCIG, there was no difference in the presence of orthostatic symptoms. Contrarily, symptomatic orthostatic hypotension was found to have an adverse effect on LCIG treatment in 3 of a total of 72 subjects (4.2%) [30]. Also, according to some other publications, orthostatic hypotension has been considered to be a possible adverse effect of LCIG treatment in 5.6–13.5% of patients [27,29,31,32,33]. The wide range of prevalence of orthostatic hypotension might be caused by the fact that most of the patients with orthostatic hypotension are asymptomatic (only one-third of these patients are symptomatic) [34,35,36], and these patients might be misdiagnosed unless screened actively.

The current controversial question is how levodopa-carbidopa preparations affect cardiovascular autonomic dysfunction in PD patients. Although orthostatic hypotension in PD occurs mainly due to the disease itself (a manifestation of cardiovascular sympathetic failure caused by degeneration of the autonomic nervous system) [10], some authors have reported that short-acting levodopa exacerbates orthostatic hypotension in PD patients [34,37,38,39]. However, according to other studies, the effect of levodopa on orthostatic hypotension has been limited [40,41,42,43]. From the pharmacological point of view, the administration of levodopa and its turnover to dopamine in the kidneys leads to increased natriuresis and decreased blood pressure [44]. Moreover, it is very important to look also at the potential effect of carbidopa as it decreases the synthesis of norepinephrine outside the brain [45], which can cause an even more robust drop in blood pressure.

We did not confirm any correlation of orthostatic hypotension symptoms with total levodopa equivalent daily dose. This may be explained by the predominant neurogenic cause of orthostatic hypotension in Parkinson’s disease (and other synucleinopathies) [46]. On the other hand, feeling light-headed after standing for some time (SCOPA-AUT 15) correlated with the functional impact of both dyskinesias and fluctuations. In addition, fainting in the past six months (SCOPA-AUT 16) correlated with the time spent with dyskinesias. Although treatment with LCIG predicted improvement in orthostatic hypotension symptoms, according to the mediation analysis, there was a significant indirect effect of a decrease in the dyskinesia and fluctuations.

Thus, we assume that continuous delivery and stable plasmatic concentrations of both levodopa and carbidopa (in LCIG) might contribute to less severe blood pressure fluctuations. Nevertheless, it is possible that LCIG treatment might improve orthostatic hypotension symptoms by several additional mechanisms, except for eliminating high peak levodopa-carbidopa blood concentration levels. Continuous dopaminergic stimulation with LCIG has a beneficial effect on overall motor performance (increased time in the “ON” state), as well as on the activities of daily living [14,25,26,27,28,29,30]. We assume that improved mobility and adjusted daily routine eliminate risk factors for orthostatic hypotension by reducing staying in bed and, thus, decreasing the accumulation of blood in the peripheral vascular system. Based on the improvement of the activities of daily living, we could also expect an enhancement of fluid intake that provides sufficient blood volume and, thus, enhances baroreflex sensitivity in PD patients on LCIG treatment. Decreased baroreflex sensitivity is significantly lower in PD patients compared to the general population, and it was in a strong positive correlation with orthostatic hypotension [47].

Last but not least, intrajejunal delivery of LCIG minimally interferes with food intake. It enables patients to have smaller portions of meals more frequently compared to patients having larger meals only between scheduled oral medication. Avoiding big portions is crucial for the prevention of postprandial hypotension, negatively affecting also orthostatic hypotension [48].

All the above-mentioned approaches are part of the nonpharmacological management of orthostatic hypotension [49], and they might be connected with better adherence in patients managed by continuous treatment methods.

As already mentioned, the results about the influence of continual LCIG infusion on blood pressure and orthostatic hypotension are contradictory. Our data support the hypothesis that continual treatment can reduce signs of “light-headedness after standing” and “fainting”. We also disclosed QoL improvement in patients after six months of LCIG treatment compared to OMT, and this change was connected with a reduction in both motor complications and symptoms of orthostatic hypotension.

Our study had some limitations. The low number of participants resulted from complicated recruitment during the COVID-19 pandemic. Another limitation was the use of subjective questionnaires without objectifying signs of orthostatic hypotension. Nevertheless, we tried to disclose a long-term impact of LCIG on orthostatic hypotension symptoms, rather than an instant effect on blood pressure levels. Our study had an open-label design; therefore double-blinded trials should be done to confirm our findings.

In summary, our study confirmed that LCIG treatment improved both motor complications and symptoms of orthostatic hypotension in patients with PD after six months of treatment. Stable plasmatic concentrations of levodopa-carbidopa with improved overall motor state and the activities of daily living might prevent significant fluctuations in blood pressure levels. Treatment with LCIG should be considered for PD patients not only with motor complications but also with severe nonmotor fluctuations responsive to dopaminergic treatment. Moreover, it can potentially reduce symptoms such as dizziness, syncope, falls, and related complications, and it results in better QoL and patient independence.

## Figures and Tables

**Figure 1 jpm-12-00718-f001:**
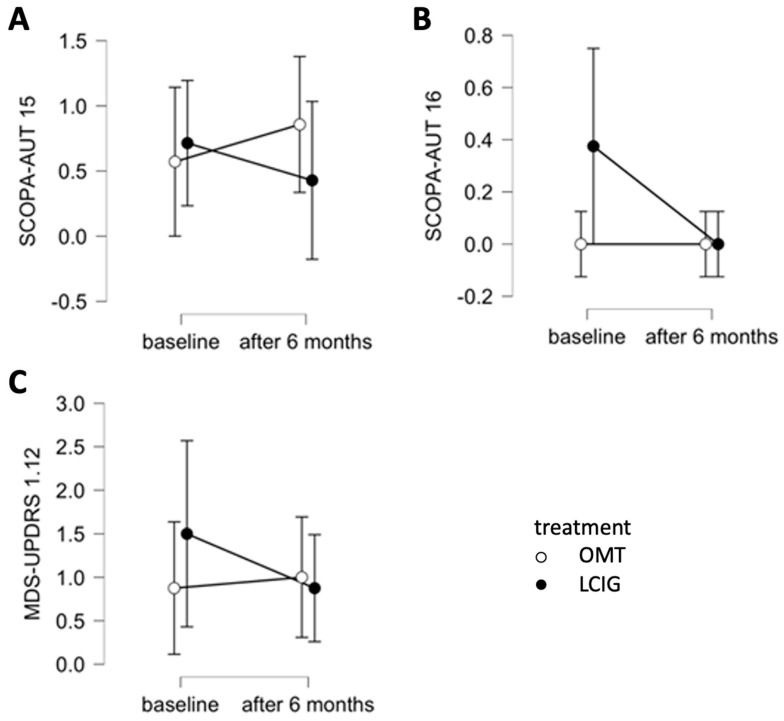
Changes after 6 months comparing LCIG group and OMT group for following parameters: (**A**) SCOPA-AUT 15 (“In the past month, did you become light-headed after standing for some time?”), (**B**) SCOPA-AUT 16 (“Have you fainted in the past 6 months?”), and (**C**) MDS-UPDRS, Part I—Nonmotor Symptoms, question 12 (“Lightheadedness on standing”).

**Figure 2 jpm-12-00718-f002:**
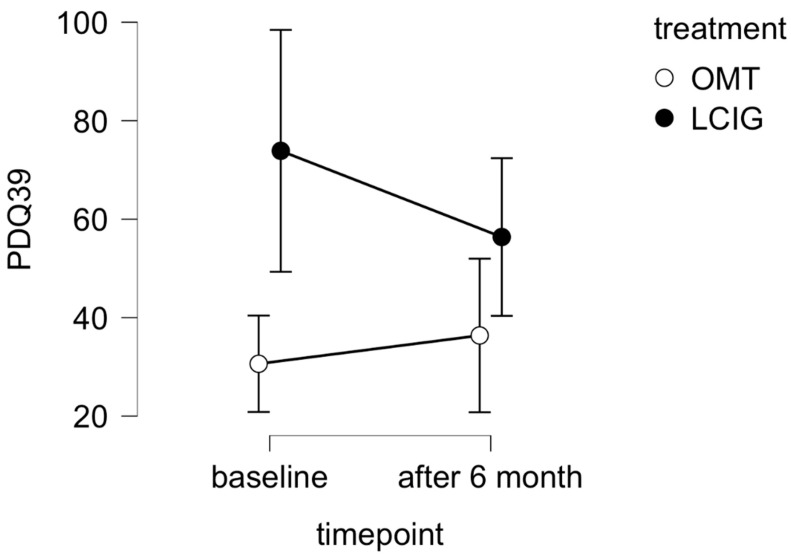
Changes in PDQ-39 total score after 6 months comparing LCIG group and OMT group.

**Figure 3 jpm-12-00718-f003:**
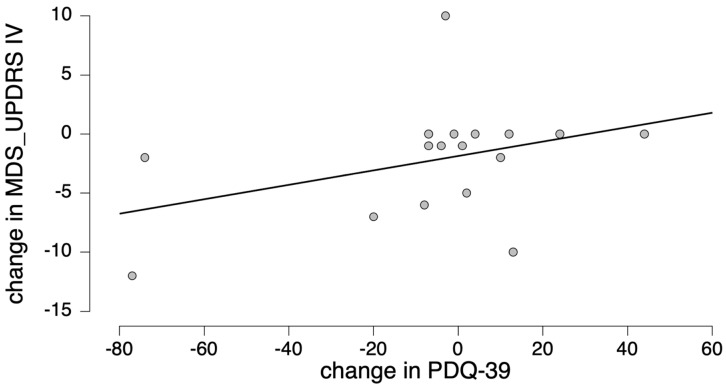
Correlation between the change in the quality of life according to the PDQ39 scale and fluctuation improvement measured by the change in MDS-UPDRS Part IV score (r_s_ = 0.554, *p* = 0.032).

**Table 1 jpm-12-00718-t001:** Descriptive statistics of basic demographic and clinical data.

	Treatment	Mean	SD	*p*-Value
Age (years)	OMT	66.125	5.410	0.358
	LCIG	68.444	5.897	
PD duration (years)	OMT	11.125	4.422	0.697
	LCIG	10.333	1.936	
LEDD (mg per day)	OMT	1323.625	622.374	0.048
	LCIG	1801.278	446.868	
DA-LEDD (mg per day)	OMT	195.3	163.7	0.248
	LCIG	109.1	129.6	

PD—Parkinson’s disease, LEDD—levodopa equivalent daily dose, DA-LEDD—levodopa equivalent daily dose of dopamine agonists, OMT—optimized medical treatment, LCIG—levodopa-carbidopa intestinal gel.

**Table 2 jpm-12-00718-t002:** Changes in parameters of symptoms of orthostatic hypotension between baseline and after 6 months in both LCIG and OMT groups (repeated measure ANOVA—small effect η^2^ < 0.01, medium effect η^2^ < 0.06, and large effect η^2^ < 0.14.

		Baseline		After 6 Months		Effect of Treatment		
Parameter	Group	Mean	SD	Mean	SD	F	*p*	η^2^
SCOPA-AUT total	LCIG	26.00	11.964	18.63	5.655	1.385	0.278	0.041
	OMT	19.25	8.396	17.88	6.958			
SCOPA-AUT 15	LCIG	0.714	0.756	0.429	0.535	8.000	0.030	0.082
	OMT	0.571	1.134	0.857	1.069			
SCOPA-AUT 16	LCIG	0.375	0.518	0.000	0.000	4.200	0.080	0.125
	OMT	0.000	0.000	0.000	0.000			
MDS-UPDRS I 1.12	LCIG	1.500	1.069	0.875	1.126	3.316	0.111	0.052
	OMT	0.875	1.126	1.000	1.069			

MDS-UPDRS I—MDS-Unified Parkinson’s Disease Rating Scale—Part I: Nonmotor symptoms; SCOPA-AUT—Scales for Outcomes in Parkinson’s Disease—Autonomic Dysfunction; F—F-ratio; *p*—*p*-value; η^2^—eta squared.

**Table 3 jpm-12-00718-t003:** Changes in parameters of MDS-UPDRS IV (motor complications) between baseline and after 6 months in both LCIG and OMT groups (repeated measure ANOVA—small effect η^2^ < 0.01, medium effect η^2^ < 0.06, and large effect η^2^ < 0.14.

		Baseline		After 6 Months		Effect of Treatment		
Parameter	Therapy	Mean	SD	Mean	SD	F	*p*	η^2^
MDS-UPDRS IV total	LCIG	9.625	5.878	5.250	2.493	10.795	0.013	0.123
	OMT	3.250	2.816	4.250	3.576			
MDS-UPDRS IV 1	LCIG	1.000	0.756	0.625	0,744	4.200	0.080	0.030
	OMT	0.250	0.463	0.250	0.463			
MDS-UPDRS IV 2	LCIG	2.000	1.512	0.750	0.886	9.211	0.019	0.093
	OMT	0.250	0.463	0.250	0.463			
MDS-UPDRS IV 3	LCIG	1.250	0.707	0.875	0.354	2.333	0.170	0.056
	OMT	0.750	0.707	0.875	0.641			
MDS-UPDRS IV 4	LCIG	2.875	1.553	1.375	0.744	8.615	0.022	0.172
	OMT	0.875	0.835	1.375	1.302			
MDS-UPDRS IV 5	LCIG	1.750	1.282	1.125	0.641	3.111	0.121	0.063
	OMT	0.875	0.991	1.250	1.282			
MDS-UPDRS IV 6	LCIG	0.750	1.488	0.500	0.756	0.368	0.563	0.010
	OMT	0.250	0.463	0.250	0.463			

MDS-UPDRS IV—MDS-Unified Parkinson’s Disease Rating Scale—Part IV: Motor complications; F—F-ratio; *p*—*p*-value; η^2^—eta squared.

## Data Availability

The data presented in this study are available upon request from the corresponding author. The data are not publicly available because the database contains patient personal data.

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
