# Peer review of "Levodopa-Carbidopa Intestinal Gel Improves Symptoms of Orthostatic Hypotension in Patients with Parkinson’s Disease—Prospective Pilot Interventional Study"

_jpm, 2022, doi:10.3390/jpm12050718_

Round 1

Reviewer 1 Report

Orthostatic hypotension in Parkinson's disease is a very important problem, and it is important to consider LCIG.
However, the number of cases in this study is too small.
The number of cases must be increased for any reason to publish a paper.

Author Response

Thank you for your comments, we agree that the number of patients is low. According to the power analysis, the number was determined to be 12 in each group, but to strict COVID restrictions in our hospital, the recruitment was difficult and we managed to enrol 17 instead of 24 patients. 

According to both your and the other reviewers' comments, methods, results and conclusions will be explained in more detail. 

From the formal point of view, the manuscript will be re-edited for English by another native speaker.

Reviewer 2 Report

This study evaluates the effects of LCIG therapy on orthostatic hypotension compared to optimized medical treatment in PD patients. The topic of this research article is very interesting, however the complexity of the disease and the small sample size make it difficult to objectively verify the real effect that LCIG has on OH.

Review: The study should elaborate in detail the meaning of OMT (optimised medical treatment): medication and doses - what kind of agonists and in what dose did the patients get with OH as a potential side effect

- those patients who received LCIG, were only on monotherapy or did they receive dopamine agonists with potential side effects such as OH?

The study should disclose the selection criteria for the patients. Were these consecutive patients, or not? Did they exclude the patients, who might also have cardiovascular comorbidities, which might lead to fainting? Patient selection should be more detailed.

It should be discussed whether the patients had other medications, that could influence OH.

Regarding the correlation between the risk of fainting and the time spent with dyskinesias: it should be highlighted whether the severity of dyskinesias can be a potential cause of falling.

Many of the cited references are more than 5 years old. This should be improved.

The sample size should be increased if possible.

There should be an approximately equal distribution between male and female patients in the study, if possible.

Author Response

Thank you for your comments.

From the formal point of view, the manuscript will be re-edited for English by another native speaker.

Responses:

The study should elaborate in detail the meaning of OMT (optimised medical treatment): medication and doses - what kind of agonists and in what dose did the patients get with OH as a potential side effect

- those patients who received LCIG, were only on monotherapy or did they receive dopamine agonists with potential side effects such as OH?

  • added to manuscript: 

    Optimized medical treatment (OMT) included any combination of oral preparations of levodopa (with carbidopa or benserazide), entacapone, rasagiline, amantadine, pramipexol and ropinirole, and rotigotine in transdermal patch. Baseline total levodopa equivalent daily dose was higher in LCIG-group (p=0.048), equivalent daily dose of dopamine agonists was not significantly different between LCIG- and OMT-group (0.248). In LCIG-group, 44% (4 out of 9) patients were treated with dopamine agonists (DA - 3 with rotigotine, 1 with pramipexol); after switching to LCIG, 5 patients were on monotherapy with LCIG and 4 patients had co-medication with rotigotine. In OMT group, 75% (6 out of 8) patients were treated by DA – 3 with pramipexol, 3 with ropinirole, this medication did not change during the follow-up.

The study should disclose the selection criteria for the patients. Were these consecutive patients, or not? Did they exclude the patients, who might also have cardiovascular comorbidities, which might lead to fainting? Patient selection should be more detailed.

added to manuscript: 

Inclusion criteria

- idiopathic PD according to the MDS Clinical Diagnostic Criteria for PD [[20]],

- willingness to participate in six months follow-up,

- ability to complete all questionnaires (assistance of caregiver was allowed).

Exclusion criteria

- Montreal cognitive assessment (MOCA) below 10 (severe dementia)

- severe cardiovascular comorbidity other than arterial hypertension.

We enrolled:

1) nine (n=9) consecutive patients in advanced stage PD according to the Delphi expert consensus panel “5-2-1 criteria”  -  oral levodopa at least five times per day, having at least 2 hours of the day with ‘Off’ symptoms or at least 1 hour of the day with troublesome dyskinesia [21];  in which oral regimens no longer adequately managed PD symptoms and were indicated for continuous LCIG treatment (LCIG-group), and

2) eight patients with idiopathic PD not fulfilling 5-2-1 criteria, treated with OMT, and matched by age, gender and PD duration with patients indicated for LCIG (OMT-group).

It should be discussed whether the patients had other medications, that could influence OH.

added to manuscript: 

In both groups, four patients were treated by antihypertensive drugs with stable dose for four weeks prior to evaluations.

Regarding the correlation between the risk of fainting and the time spent with dyskinesias: it should be highlighted whether the severity of dyskinesias can be a potential cause of falling.

all the questionnaires were administrated by trained investigators who instructed patients to distinguish falls from syncope/fainting

Many of the cited references are more than 5 years old. This should be improved.

we were surprised that there is not much recent publications about treatment options of OH in PD (there are more about pathophysiology or epidemiology)

The sample size should be increased if possible.

We agree that the number of patients is low. According to the power analysis, the number was determined to be 12 in each group, but to strict COVID restrictions in our hospital, the recruitment was difficult and we managed to enrol 17 instead of 24 patients. 

There should be an approximately equal distribution between male and female patients in the study, if possible.

Parkinson's disease is more prevalent in male gender, anywayswe admit that our male preponderance is high. 

Reviewer 3 Report

The paper “Levodopa-carbidopa intestinal gel improves symptoms of orthostatic hypotension in patients with Parkinson’s disease - prospective interventional study” studies the effects of Levodopa-carbidopa intestinal gel on patient perception about orthostatic hypotension. The paper can be a challenge to readers that are not experts in clinical practice involving Parkinson’s disease patients, because it contains a lot of abbreviations and lack explanation of some parameters and scores.

            The paper concludes that levodopa-carbidopa intestinal gel treatment improves both complications and symptoms of orthostatic hypotension in comparison to the treatment identified as optimized medical treatment. However, the groups of levodopa-carbidopa intestinal gel and of optimized medical treatment differ in the initial values of several parameters before treatments, where the patients in the levodopa-carbidopa intestinal gel present higher values than the other group. This can create a false perception of improvement in patients treated with levodopa-carbidopa intestinal gel.

            I have a few comments for author reflection and consequent improvement of the work.

Major comments:

  1. The use of several abbreviations over the manuscript and the lack of information of several parameters that are used to evaluate the patients hinders the understanding of the paper. Moreover, the authors use very specific technical jargon without explaining its meaning. For example, the abstract that should be clear and easily understandable is enriched in abbreviations and technical jargon. The authors should try to communicate in a simple way a clear message. I also present here some specific examples:

- the authors refer that fluctuations are a side effect of long-term use of oral levodopa preparations, but they do not explain what are fluctuations. The same happens to OFF and ON state.

- there is a lack of explanation of the different parameters: What is Hoehn & Yah score and what does it indicate? What are the questions 2 and 4 that the authors refer in the section 3.2.

- which comparisons are presented in table 3 in the column entitled “difference”?

- is it required abbreviate “activities of daily living” (ADL) or orthostatic hypotension?

  1. One of the reasons why there are an apparent improvement in the group treated with the levodopa-carbidopa intestinal gel is because the group optimized medical treatment present lower values of each parameter in the beginning of the study. Why is there this difference? What were the criteria for the selection of patients for levodopa-carbidopa intestinal gel? Also specify what is the optimized medical treatment. Moreover, the authors should use a statistical test that takes into account the existence of two factors – treatment (levodopa-carbidopa intestinal gel vs group optimized medical) and time (before vs after treatment), because the currently statistical analysis ignore the initial differences between the two groups.

  1. The authors should explain in the text why this article is innovative and in what differ from the other studies with levodopa-carbidopa intestinal gel and non-motor symptoms in Parkinson’s disease. In the discussion, the authors refer several other works that studied the effects of levodopa-carbidopa intestinal gel in Parkinson’s disease, without explaining how this work differs from the previous published papers.

  1. The authors refer that orthostatic hypotension occurs in PD mainly due to the disease itself. The authors should specify the mechanisms by which orthostatic hypotension occurs in PD patients and clarify the reasons why levodopa-carbidopa intestinal gel attenuates orthostatic hypotension.

Minor

  1. The paragraphs should be revised. Each paragraph should present a specific idea. While in introduction the presence of more paragraphs will improve the organization of the ideas, in the discussion there are paragraphs that continues the idea of the previous text.

  1. The authors performed several correlations between parameters. I recommend the authors to present some graphs illustrating the most important correlations.

  1. In the beginning of page 11, the second sentence is extremely vague. There is a “statistically significant improvement” with what?

Author Response

Thank you very much for your comments.

Responses:

Major comments:

The use of several abbreviations over the manuscript and the lack of information of several parameters that are used to evaluate the patients hinders the understanding of the paper. Moreover, the authors use very specific technical jargon without explaining its meaning. For example, the abstract that should be clear and easily understandable is enriched in abbreviations and technical jargon. The authors should try to communicate in a simple way a clear message.

we tried to re-write the text in more clear manner

I also present here some specific examples:

  • the authors refer that fluctuations are a side effect of long-term use of oral levodopa preparations, but they do not explain what are fluctuations. The same happens to OFF and ON state.

added to the text: 

Advanced PD is associated with motor and non-motor fluctuations. They are characterized as diurnal changes in “ON” time (i.e. time of favourable effect of dopaminergic treatment with good motor performance) and “OFF” time (the time when PD medication wears off and has an insufficient effect on motor function). They arise as a side effect of long-term use of oral levodopa preparations [[12]] and lead to a more complicated PD management, an increased need for nursing care and impaired quality of life
  • there is a lack of explanation of the different parameters: What is Hoehn & Yah score and what does it indicate?

updated in manuscript: The median of Hoehn & Yahr score (used to describe the symptom progression or stage of PD)[22] was 2.5 for OMT-group and 3.0 for LCIG-group (IQR 0.125, and 0.000, respectively).

What are the questions 2 and 4 that the authors refer in the section 3.2.

corrected in the manuscript: question 2 - “Functional impact of dyskinesias” (rs=0.674, p=0.006), and question 4 - “Functional impact of fluctuations” - rs=0.640, p=0.010).

- which comparisons are presented in table 3 in the column entitled “difference”?

corrected in the manuscript: difference = effect of treatment - comparison of LCIG and OMT on respective parameter

- is it required abbreviate “activities of daily living” (ADL) or orthostatic hypotension?

corrected in the manuscript

One of the reasons why there are an apparent improvement in the group treated with the levodopa-carbidopa intestinal gel is because the group optimized medical treatment present lower values of each parameter in the beginning of the study. Why is there this difference?

OMT patients are matched with age and even PD duration, but the difference in baseline scores can be explained by slower progression in these patients - nevertheless, this fact was taken into consideration in our statistical analysis - please, see below

What were the criteria for the selection of patients for levodopa-carbidopa intestinal gel?

added to manuscript:

patients in advanced stage PD according to the Delphi expert consensus panel “5-2-1 criteria”  -  oral levodopa at least five times per day, having at least 2 hours of the day with ‘Off’ symptoms or at least 1 hour of the day with troublesome dyskinesia [21];  in which oral regimens no longer adequately managed PD symptoms and were indicated for continuous LCIG treatment (LCIG-group)

Also specify what is the optimized medical treatment.

added to manuscript: 

Optimized medical treatment (OMT) included any combination of oral preparations of levodopa (with carbidopa or benserazide), entacapone, rasagiline, amantadine, pramipexol and ropinirole, and rotigotine in transdermal patch – in order to achieve a satisfactory motor state.

Moreover, the authors should use a statistical test that takes into account the existence of two factors – treatment (levodopa-carbidopa intestinal gel vs group optimized medical) and time (before vs after treatment), because the currently statistical analysis ignore the initial differences between the two groups.

we used repeated-measures ANOVA, in which three relationships are taken into consideration - between-group difference, between-timepoint difference, and treatment*timepoint relationship reflecting the real effect of treatment on outcomes regardless of baseline differences (we report these coefficients)

  1. The authors should explain in the text why this article is innovative and in what differ from the other studies with levodopa-carbidopa intestinal gel and non-motor symptoms in Parkinson’s disease. In the discussion, the authors refer several other works that studied the effects of levodopa-carbidopa intestinal gel in Parkinson’s disease, without explaining how this work differs from the previous published papers.

added/modified: As already mentioned, the results about the influence of continual LCIG infusion on blood pressure and orthostatic hypotension are contradictory. Our data support the hypothesis that continual treatment can reduce signs of  “light-headedness after standing” and/or “fainting”.   We also disclosed QoL improvement in patients after 6 months of LCIG treatment compared to OMT; and this change was connected with reduction in both motor complications and symptoms of orthostatic hypotension.

  1. The authors refer that orthostatic hypotension occurs in PD mainly due to the disease itself. The authors should specify the mechanisms by which orthostatic hypotension occurs in PD patients and clarify the reasons why levodopa-carbidopa intestinal gel attenuates orthostatic hypotension.

added/corrected in the manuscript: 

Although OH in PD occurs mainly due to the disease itself (manifestation of cardiovascular sympathetic failure caused by degeneration of the autonomic nervous system) [10], some authors reported that short-acting levodopa exacerbates OH in PD patients [34], [37]–[39].   Thus, we assume that continuous delivery and stable plasmatic concentrations of both levodopa and carbidopa (in LCIG) might contribute to less severe blood pressure fluctuations.

Minor

  1. The paragraphs should be revised. Each paragraph should present a specific idea. While in introduction the presence of more paragraphs will improve the organization of the ideas, in the discussion there are paragraphs that continues the idea of the previous text.

corrected in the manuscript

  1. The authors performed several correlations between parameters. I recommend the authors to present some graphs illustrating the most important correlations.

added to manuscript

  1. In the beginning of page 11, the second sentence is extremely vague. There is a “statistically significant improvement” with what?

corrected in manuscript: According to some studies, LCIG significantly improved the score of the cardiovascular domain of the Non-Motor Symptoms Scale for Parkinson´s Disease (NMSS), which also focuses on the symptoms of OH

Round 2

Reviewer 1 Report

I believe that the number of cases should be about twice as many as the current number in order to publish a paper.

Author Response

there are more studies about the effect of LCIG in up to 2O, or up to 10 subjects

https://pubmed.ncbi.nlm.nih.gov/30515445/

https://pubmed.ncbi.nlm.nih.gov/30515445/

https://pubmed.ncbi.nlm.nih.gov/26777085/

we adjusted the name to PILOT study accordingly

Reviewer 3 Report

The authors clearly addressed the majority of my comments.  The abstract was successfully clarified, and the technical jargon was in general explained. The authors also described the inclusion criteria of the patients in each condition and the treatments performed by the patients.

The authors added a new figure, however I could not find this figure in the files that I had access, so I cannot give my opinion about it.

Please, see my minor comments:

  • I recommend removing from the abstract the statistical parameters to simplify the reading.
  • The use and the identification of the abbreviations were improved, although some abbreviations continue unnecessary – for example blood pressure (BP). Note that the abbreviation Qof appears in the second sentence of introduction and its meaning only appears in the beginning of the second page of the introduction.
  • The authors introduced in Table 1 the parameter DA-LEDD without explaining it.
  • The authors should explain in the manuscript how the differences between treatments identified in tables 2 and 3 were calculated.
  • Note that the numbers of references over the text are inside between 2 double brackets [[]].

Author Response

The authors clearly addressed the majority of my comments.  The abstract was successfully clarified, and the technical jargon was in general explained. The authors also described the inclusion criteria of the patients in each condition and the treatments performed by the patients.

The authors added a new figure, however I could not find this figure in the files that I had access, so I cannot give my opinion about it.

figures added also to the manuscript

Please, see my minor comments:

  • I recommend removing from the abstract the statistical parameters to simplify the reading.

removed

  • The use and the identification of the abbreviations were improved, although some abbreviations continue unnecessary – for example blood pressure (BP).

changed all of them

  • Note that the abbreviation Qof appears in the second sentence of introduction and its meaning only appears in the beginning of the second page of the introduction.

corrected

  • The authors introduced in Table 1 the parameter DA-LEDD without explaining it.

explained

  • The authors should explain in the manuscript how the differences between treatments identified in tables 2 and 3 were calculated.

added to the Methods section and added also to Tables

  • Note that the numbers of references over the text are inside between 2 double brackets [[]].

corrected